# Effects of Titanium Dioxide Nanoparticles on Cell Growth and Migration of A549 Cells under Simulated Microgravity

**DOI:** 10.3390/nano12111879

**Published:** 2022-05-31

**Authors:** Mei Wang, Jinxia Li, Shunyu Zhang, Yue You, Xianyu Zhu, Huandong Xiang, Liang Yan, Feng Zhao, Yunhui Li

**Affiliations:** 1CAS Key Lab for Biomedical Effects of Nanomaterials and Nanosafety, Institute of High Energy Physics, Chinese Academy of Sciences, Beijing 100049, China; wmjude2018@163.com (M.W.); lijinxia@ihep.ac.cn (J.L.); syzhang@ihep.ac.cn (S.Z.); youyue@ihep.ac.cn (Y.Y.); zhuxianyu@ihep.ac.cn (X.Z.); xianghd@ihep.ac.cn (H.X.); yanliang@ihep.ac.cn (L.Y.); 2Key Laboratory of Environmental Medicine Engineering of Ministry of Education, School of Public Health, Southeast University, Nanjing 210000, China

**Keywords:** simulated microgravity, titanium dioxide nanoparticles, cell migration, cytoskeleton, focal adhesion kinase

## Abstract

With the increasing application of nanomaterials in aerospace technology, the long-term space exposure to nanomaterials especially in the space full of radiation coupled with microgravity condition has aroused great health concerns of the astronauts. However, few studies have been conducted to assess these effects, which are crucial for seeking the possible intervention strategy. Herein, using a random positioning machine (RPM) to simulate microgravity, we investigated the behaviors of cells under simulated microgravity and also evaluated the possible toxicity of titanium dioxide nanoparticles (TiO_2_ NPs), a multifunctional nanomaterial with potential application in aerospace. Pulmonary epithelial cells A549 were exposed to normal gravity (1 g) and simulated gravity (~10^−3^ g), respectively. The results showed that simulated microgravity had no significant effect on the viability of A549 cells as compared with normal gravity within 48 h. The effects of TiO_2_ NPs exposure on cell viability and apoptosis were marginal with only a slightly decrease in cell viability and a subtle increase in apoptosis rate observed at a high concentration of TiO_2_ NPs (100 μg/mL). However, it was observed that the exposure to simulated microgravity could obviously reduce A549 cell migration compared with normal gravity. The disruption of F-actin network and the deactivation of FAK (Tyr397) might be responsible for the impaired mobility of simulated microgravity-exposed A549 cells. TiO_2_ NPs exposure inhibited cell migration under two different gravity conditions, but to different degrees, with a milder inhibition under simulated microgravity. Meanwhile, it was found that A549 cells internalized more TiO_2_ NPs under normal gravity than simulated microgravity, which may account for the lower cytotoxicity and the lighter inhibition of cell migration induced by the same exposure concentration of TiO_2_ NPs under simulated microgravity at least partially. Our study has provided some tentative information on the effects of TiO_2_ NPs exposure on cell behaviors under simulated microgravity.

## 1. Introduction

Nowadays, nanotechnology has made remarkable progress in the fields of environment, medicine, engineering, biology, communication and materials [1,2]. In recent years, all countries have been actively promoting the development of nanotechnology in the field of space with a focus on the research of novel exploitable nanomaterials [3]. The use of nanomaterials will permeate all aspects of space in the future. For example, titanium dioxide nanoparticles (TiO_2_ NPs) can be used as a strengthening agent of metals and alloys in aircraft manufacturing, which can improve the mechanical properties and wear resistance of materials and have a heat insulation effect [4,5,6]. Graphene oxide (GO) nanocomposites, due to their sensitivity to humidity, have great potential to be used in the humidity sensing system of space suits [7]. However, the wide application of nanomaterials increases the frequency of the exposure of astronauts to nanomaterials in this confined microgravity environment, raising the great health concerns for astronauts in the extreme condition. Thus, it is of great significance to evaluate the impacts of nanomaterials on human health under microgravity.

There have been some reports on the health effects of microgravity on astronauts in the space environment. Microgravity has been reported to affect multiple organs and tissues of astronauts [8], with effects including cardiac function decline, cardiac atrophy and postural hypotension [9,10,11], leukocytosis [12], osteoporosis, decreased bone density and muscle atrophy [13,14], transient changes in lung muscles [15,16] and even psychological problems [17].

Real microgravity can be achieved by drop towers, parabolic flights, rockets, spaceships or space labs on the International Space Station (ISS) [18]. However, since space research missions are costly and time-consuming, there are few opportunities to carry out space biology experiments. In addition, the short duration of microgravity conditions obtained using drop towers, parabolic flight or rockets, usually measured in seconds or minutes, poses a great limitation on the studies of many complex and long-lasting biological processes [18,19]. Therefore, it is essential to establish an acceptable model to simulate microgravity on the ground. Currently, microgravity simulators specifically used for cell research principally include the suspension induced by strong magnetic fields, two-dimension and three-dimensional clinostats, rotating wall vessels (RWV) and random positioning machines (RPM) [18,20,21,22]. Among them, the RPMs have been considered to be an efficient tool to construct the condition of simulated microgravity [22,23,24]. Depending on the constant rotation to change the direction of the sample relative to gravity, the sum of the gravity vector is zero during one rotation period, so that the sample does not "feel" the presence of gravity. The range of acceleration remains around the 10^−3^ g range, which is considered to represent microgravity conditions [18].

To date, many researchers have studied the behaviors of normal cells and tumor cells including proliferation, apoptosis, migration, and cytoskeleton or focal adhesion under real and simulated microgravity [25,26,27,28,29,30,31]. Firstly, microgravity may affect cell proliferation. However, based on the currently available studies, different conclusions exist on whether microgravity can promote or inhibit cell proliferation. For instance, human pluripotent stem cells-derived microscale progenitor cardiac spheres were exposed to simulated microgravity using an RPM for 3 days during their differentiation to cardiomyocytes, and it was shown that simulated microgravity could promote the differentiation of pluripotent stem cells to cardiomyocytes with high viability [27]. Ahn et al. found that simulated microgravity could promote the proliferation and migration of A549 and H1703 cells with the up-regulation of migration-related genes [25]. The adverse effect of simulated microgravity on this aspect has also been reported. The exposure of lung cancer cells to simulated microgravity using an RPM might induce apoptosis and affect the alignment of actin filaments [29]. Simulated microgravity was also reported to alter the expression of some miRNAs in human vascular endothelial cells (HUVECs), some of which were verified to be associated with apoptosis [30]. A comparative study between real and simulated microgravity was conducted to explore the effects on human bone marrow monocyte cell line U937, and a decrease in proliferation rate was observed in both real and simulated microgravity [32,33]. Shi et al. [34] found that the 12-day microgravity environment inhibited the differentiation and growth of mouse bone marrow-derived macrophages through RAS/ERK/NFκB and metabolic pathways, both in spaceflight (SJ-10) and ground simulated microgravity. Prasad et al. [35] in their review, found that most tumor cell lines showed increased apoptosis when exposed to real or simulated microgravity, possibly due to the abnormal expression of apoptosis-related proteins and tumor suppressor genes.

Many studies have focused on the changes in the cytoskeleton in real microgravity. Rijken et al. [36] observed a higher filamentous actin density and reduced stress fiber structure in A431 cells in sounding rocket experiments. Subsequently, changes in the actin cytoskeleton in osteoblasts [37] (4 days) and monocytes [38] (1 day) were studied and a reduction in the number of stress fibers was found. Microgravity may impact on cellular morphology and functions by disrupting cytoskeleton and adhesion structures [39,40]. Structural changes in actin networks have been reported in experiments under simulated microgravity using RPM and RWV, including a reduction in the width and number of stress fibers [41,42], decrease in actin density [43,44], and cell border localization [45], which confirmed the results of various observations in spaceflight experiments. However, other studies have suggested the opposite conclusion, showing that the actin network of Jurkat cells [46] and cardiomyocytes [47] did not change in microgravity.

Although nanotechnology has been extensively explored for medical purposes, its application in the space environment is still in its infancy, with little retrievable literature. Some studies have shown that nanomaterials can reverse the adverse effects of microgravity on cells and organisms. For instance, nanoceria will be applied to rescue the oxidative stress of skeletal muscles induced by microgravity and cosmic radiation, suggesting the promising applications of nanomaterials in harsh environments of low Earth orbit and deep space [48]. Furthermore, a team identified the role of mitochondrial uncoupling protein 2 (Ucp2) in mediating the cellular responses to space [49]. Other studies have suspended the tails of mice to simulate microgravity exposure and found that reactive mesoporous silica nanoparticles (LE@MSNs-CYC) might fight anxiety and improve the learning and memory of the suspended mice [50]. Nanoparticles and osteoporosis (NATO) projects were the first to investigate the potential of calcium- and strontium-containing hydroxyapatite nanoparticles on bone remodeling of human bone marrow mesenchymal stem cells under normal gravity (1 g), RPM, and ISS. The results showed the promise of nanomaterial as a component of bone replacement synthetic materials for pharmaceutical preparations or food supplements [51].

TiO_2_ NPs are extensively studied multifunctional nanomaterial with attractive applications in aerospace engineering. TiO_2_ NPs have self-cleaning [52], ultraviolet protection [53], and antibacterial properties [54], attractive for the application in space suits. However, their health hazards have also been suggested such as acute lung reactions, and heart and kidney damage in mice [55,56,57], have also been suggested, with the involvement of cell membrane damage, cell cycle disruption, growth inhibition, the release of inflammatory factors and oxidative stress [58,59,60,61]. Therefore, the possible inhalation of TiO_2_ NPs by astronauts during space missions might affect their health especially in the extreme conditions of microgravity and radiation. Our present study aims to explore the effect of TiO_2_ NPs on the behaviors of A549 cells in a simulated microgravity environment. The A549 cell line is often used as an alveolar epithelial cell model for evaluating nanomaterial toxicology. In this experiment, an RPM was used to simulate ground-based microgravity. The effects of TiO_2_ NPs on the morphology, growth, proliferation, adhesion and migration of A549 cells under simulated microgravity were evaluated.

## 2. Materials and Methods

### 2.1. Nanoparticle Characterization 

TiO_2_ NPs were purchased from Shanghai Macklin Biochemical Co., Ltd. (Shanghai, China). The size and morphology of TiO_2_ NPs were observed using a transmission electron microscope (TEM) (JEM2100Plus) (JEOL, Tokyo, Japan). The elemental compositions were analyzed using an energy-dispersive spectrometer (EDS) (JEOL, Tokyo, Japan) attached to the TEM. The sample for TEM analysis was prepared as follow: TiO_2_ NPs were suspended in ultra-pure water to obtain a dispersion of 50 μg/mL. After sonication for 30 min (100 W), the dispersion was loaded onto an ultrathin carbon film on a Cu grid, and dried in low vacuum at room temperature.

Size distribution and zeta potential of the TiO_2_ NPs dispersion were determined using dynamic light scattering (DLS) (NanoBrook Omni, Holtsville, NY, USA). The average particle size, polydispersity index (PDI) and zeta potential of 50 μg/mL TiO_2_ NPs dispersion in ultrapure water were represented as average ± standard deviation of three different measurements, each with 15 readings.

X-ray diffraction (XRD) analysis of TiO_2_ NPs sample was performed using a D8 Advance X-ray diffractometer (Bruker, Karlsruhe, Germany). The diffraction pattern was collected in Cu Kα radiation over an angular range (20−80°), with a step size of 0.02°. The measurement was conducted at room temperature using a zero-diffraction silicon substrate. 

The formation of TiO_2_ NPs in the powder form was recorded using Fourier transform infrared (FT-IR) spectrometer (Thermo Scientific, Waltham, MA, USA) coupled with a Nicolet iN10 MX spectrograph.

### 2.2. Cell Culture

The human alveolar epithelial cell line A549 was purchased from American Type Culture Collection (ATCC). Cells were maintained in DMEM (Hyclone Laboratories, Logan, UT, USA) supplemented with 10% (*v*/*v*) fetal bovine serum (Gibco Invitrogen, Grand Island, NY, USA) and 1% penicillin/streptomycin (Gibco Invitrogen, Grand Island, NY, USA) at 37 °C, in a 5% CO_2_ humid atmosphere. The medium was changed every 2 days until the cultured cells reached approximately 80% confluence.

### 2.3. Simulated Microgravity and Cell Exposure 

In this experiment, the dual-axis drive (3D) clinostat (SM-31), developed by Technology and Engineering Center for Space Utilization, Chinese Academy of Sciences, was used to simulate microgravity. During the experiment, the internal and external frame speeds of the controller of the dual-axis drive clinostat were set at random to investigate the influence of simulated microgravity on the ground [62]. Rotation causes gravity vector unrecognizable to cells. The RPM was sterilized via UV light and 75% ethanol before being put in a CO_2_ incubator.

The cell culture dishes were placed into the RPM to simulate the microgravity exposure. All the dishes were filled with the culture medium without bubbles to avoid the impact of shear force on cells. According to the experimental requirements, the samples were rotated around the dual-axis at 6 rpm [31,63], and the rotation time was 48 h. The rotation speed (6 rpm) helped construct a simulated microgravity condition with negligible response of cells to the change in the rotational direction (µg group). Meanwhile, the cells cultured without rotation were considered as the normal gravity group (g group).

### 2.4. Cell Viability Assay

Cell viability was tested using the CCK-8 assay (Dojindo Laboratories, Tokyo, Japan). A549 cells were seeded into 35 mm petri dishes (2 × 10^4^ cells/well) and cultured overnight at 37 °C. The 35 mm petri dishes were partitioned into four sections with equal areas to allow four replicates in each group simultaneously. Then, cells were incubated with a fresh medium containing different concentrations of TiO_2_ NPs (0, 25 and 100 µg/mL) for 48 h under normal gravity or simulated microgravity, respectively. At the end of incubation, cells were washed with PBS and 500 µL of 10% CCK-8 solution was added to the cells for a further incubation of 40 min. Afterward, the supernatants in each well were collected in a 1.5 mL centrifuge tube. After centrifugation for 5 min (12,000 rpm), 80 µL of supernatant (3 replicates per tube) was transferred to 96-well plates. Absorbance was measured in each well at 450 nm using a microplate reader (Thermo Scientific, Waltham, MA, USA).

### 2.5. Annexin V-FITC/PI Double Staining

Annexin V-FITC/PI assay kit (Dojindo Laboratories, Tokyo, Japan) was used to analyze apoptosis. Cells were cultured in 35 mm culture dishes (2 × 10^5^ cells/dish). After attachment, the well-grown cells were incubated with TiO_2_ NPs (0, 25 and 100 µg/mL) for 48 h under simulated microgravity or normal gravity. Then, all the cells (both the attached and floating cells) were collected by trypsinization and centrifugation. The obtained cells were washed twice with PBS, resuspended in 500 µL of binding buffer containing 5 µL of Annexin V-FITC and 5 µL of PI, then incubated for 15 min in the dark at room temperature. Apoptosis was immediately analyzed using flow cytometry (BD LSRFortessa, USA). The images were then processed using Flow Jo software (version 10, 2021, BD, New York, NY, USA). Three independent experiments in duplicate were performed.

### 2.6. F-actin Staining

F-actin can be visualized by TRITC-phalloidin staining via confocal laser microscopy. In brief, cells (1 × 10^5^ cells/dish) were treated with TiO_2_ NPs (0, 25 and 100 µg/mL) for 48 h under simulated microgravity and normal gravity, respectively. At the end of incubation, cells were washed twice with PBS, fixed with 4% paraformaldehyde for 10 min at room temperature. Then cells were permeabilized with 0.5% Triton X-100 for 5 min followed by TRITC-phalloidin staining (100 nM, Solarbio, Beijing, China, Cat: CA1610, diluted with PBS) at dark room temperature for 30 min. Then the samples were loaded onto STEDYCON/STED ultra-high resolution confocal laser microscopy imaging system for F-actin observation (Stedycon, Abberior, Göttingen, Germany) and the representative fluorescent images were captured.

### 2.7. Wound Healing Assay 

Cell migration ability can be evaluated by in vitro wound healing assay. In brief, cells were seeded into culture dishes (2 × 10^5^ cells/dish) and cultured till 90% confluence. Then a 10 µL pipette head was used to draw a line on the culture dish every 0.5~1 cm, and cells were washed with PBS for 3 times to remove any floating cells. The scratches (straight-line gap) at 0 h were observed to ensure the consistency of the scratches under an optical microscope (IX73, Olympus, Tokyo, Japan).Then, cells were incubated with serum-free medium containing different concentrations of TiO_2_ NPs (0, 25 and 100 μg/mL) for 48 h and the scratches were observed again. To observe the changes of scratches more explicitly, cells were fixed with 4% paraformaldehyde and stained with crystal violet for 15 min. Afterwards, 5 fields of scratches in each group were randomly selected and photographed again. The mean percentage of residual open area was compared with the cell-free surface counterpart in each group in T0 using ImageJ software.

### 2.8. Immunofluorescence Analysis 

In the present study, immunofluorescence analysis was used to visualize the proteins of vinculin, E-cadherin and integrin β1 in cells of different treatment groups. In brief, cells were seeded into the confocal dishes (1 × 10^5^ cells/dish) and cultured untill 50% confluence. Then the cells were exposed to different concentrations of TiO_2_ NPs dispersions (0, 25 and 100 μg/mL) under simulated microgravity or normal gravity for 48 h. At the end of incubation, cells were fixed with 4% paraformaldehyde for 10 min, permeabilized in 0.1% Triton X-100 for 10 min and then blocked with PBS (1% BSA ) (Sigma, St. Louis, MO, USA) for 1 h. Between each step, cells were washed with PBS three times. Then, cells were incubated overnight at 4 °C with the primary antibody against anti-vinculin (1:100), anti-E-cadherin (1:200), and anti-integrin β1 (1:200), respectively, followed by the incubation with FITC-conjugated goat anti-rabbit IgG (1:500, Beyotime Biotechnology, Shanghai, China) for 1 h at room temperature. Finally, after nuclei (blue) staining with Hoechst 33342 (Beyotime Biotechnology, Shanghai, China), cells were loaded onto the confocal laser scanning microscope (Nikon A1, Tokyo, Japan) and the representative fluorescent images of cells were taken.

### 2.9. Western Blot Analysis 

A549 cells were seeded into culture dishes. After adherence, cells were exposed to different concentrations of TiO_2_ NPs dispersions under simulated microgravity or normal gravity for 48 h. As the incubation time terminated, cells were washed twice with ice-cold PBS and cell pellets were lysed with RIPA buffer (C1053, Applygen, Beijing, China) containing Protease Inhibitor Cocktail (P1266, Applygen, Beijing, China) and phenyl methyl sulfonyl fluoride (PMSF, 1 mM). Then, cell extracts were centrifuged at 12,000 rpm for 15 min at 4 °C. The protein content of supernatant fractions was determined by using the Bicinchoninic Acid Assay (BCA) protein detection kit (Beyotime Biotechnology, Beijing, China). For immunoblot analysis, cell extracts (20 µg of protein for each sample) were separated on SDS-PAGE gels. Proteins were transferred onto 0.22 µm polyvinylidene difluoride (PVDF) membranes (Millipore, Boston, MA, USA). Then membranes were blocked in 5% bovine serum albumin (BSA) for 1 h at room temperature and incubated overnight at 4 °C with primary antibody, followed by 1 h incubation of HRP-conjugated secondary antibody (anti-rabbit IgG, HRP-linked antibody, Cell Signaling Technology, Cat#7074P2) at room temperature. Finally, the bands of target proteins were visualized using ECL-Plus according to the manufacturer’s instructions and immunoreactive bands were captured using a chemiluminescence imaging system (Azure C300, Azure Biosystems, Dublin, CA, USA). GAPDH was used as the loading protein. The relative expression values of target proteins were normalized to the amount of GAPDH. Amounts of proteins were assessed by semi-quantitative ImageJ software analysis (Version 1.8.0, 2021, National Institutes of Health, Bethesda, MD, USA). Each experiment was performed at least three times independently.

### 2.10. Inductively Coupled Plasma-Optical Emission Spectrometer (ICP-OES)

Cellular uptake of TiO_2_ NPs was determined by measuring intracellular titanium concentration using inductively coupled plasma-optical emission spectrometry (ICP-OES). Briefly, 2 × 10^5^ cells/well were seeded in culture dishes. After adhesion, cells were exposed to TiO_2_ NPs (0, 25 and 100 μg/mL) under normal gravity and simulated microgravity. After 48 h, cells were washed with PBS, and collected by trypsinization. Cell count was conducted using trypan blue staining. Then cells were suspended in 100 μL PBS and stored at 4 °C until digestion. Samples were transferred to a quartz beaker, then 3 mL of ultrapure HNO_3_ was added and the samples were heated for 2 h at 120 °C. Afterwards, 1 mL H_2_O_2_ was added for further heating digestion, and then the samples were dried by heating at 180 °C. Next, the obtained solutions (about 200 μL) were transferred to a polypropylene vessel and diluted to 3 mL with 2% HNO_3_. A calibration curve was constructed using standard solutions prepared by serial dilution of a reference solution (NIM-RM3041, National Institute of Metrology, 100 mg/L) and intracellular Ti concentrations of the samples were measured with a Optima 8000 spectrometer (PerkinElmer, Waltham, MA, USA). Cellular uptake of TiO_2_ NPs was calculated against the cell counts (µg/10^5^ cells).

### 2.11. Statistical Analysis

Statistical analysis was performed with two-way ANOVA (GraphPad Prism 7.00) followed by Tukey′s multiple comparisons test. All statistical parameters were presented as mean ± standard deviation (SD). *P* < 0.05 was considered significant.

## 3. Results

### 3.1. Physicochemical Characterization of TiO_2_ NPs

The representative transmission electron microscopy (TEM) image of TiO_2_ NPs showed a relatively uniform structure (Figure 1A) with the particle size of 18-24 nm (Figure 1C). The high-resolution transmission electron microscopy image showed that the crystal lattice of TiO_2_ NPs is 0.35 nm, corresponding to 101 crystal planes (Figure 1B). The corresponding EDS results confirmed the presence of Ti and O in the tested sample, suggesting that TiO_2_ NPs sample was relatively pure (Figure 1C). The results from dynamic light scattering (DLS) analysis revealed that the average hydrodynamic diameter (HD) of TiO_2_ dispersed in ultrapure water was 122.11 ± 10.33 nm with the polydispersity index (PDI) about 0.15 ± 0.09 (Figure 1D). Zeta potential was tested to be −21.97 ± 2.57 mV, indicating a good dispersion stability of TiO_2_ NP dispersion. A very strong diffraction peak at 2θ = 25.3 could be seen, indicating the (101) planes in XRD patterns of TiO_2_ (Figure 1E), indexed as the anatase crystalline phase (JCPDS 84-1285). The well-defined peaks verified TiO_2_ NPs with high quality.

Additionally, the functional group analysis of TiO_2_ NPs using an FT-IR spectrophotometer characterized hydroxy–OH groups at 1625 and 3413 cm^−1^, which represent flexural and stretching vibration, respectively (Figure 1E). And the broad bands at 725 cm^−1^ was corresponding to the stretching vibration of Ti-O and Ti-O-Ti (Figure 1F). Due to the large polarity of Ti-O bond in TiO_2_ NPs, the water adsorbed on the surface was dissociated, which formed hydroxyl group easily. The surface hydroxyl group can improve the performance of TiO_2_ NPs as adsorbent and various carriers, also providing great flexibility for surface modification.

### 3.2. Cytotoxicity Evaluation of TiO_2_ NPs Exposure under Simulated Microgravity and Normal Gravity

First, the cytotoxicity of TiO_2_ NPs exposure on A549 cells under normal gravity was determined and the results showed that no obvious cytotoxicity was induced by TiO_2_ NPs exposure within 24 h (Figure 2A), consistently with the reported studies [61]. Then, we extended the observation period to 48 h. As demonstrated in Figure 2B, at 24 h of TiO_2_ NPs exposure, no obvious increase in cell viability was detected both under normal gravity and simulated microgravity. However, when the exposure time was extended to 48 h, the viability of 100 µg/mL TiO_2_-exposed cells was decreased to 87.96% of the viability of TiO_2_-unexposed cells under simulated microgravity (Figure 2C). Similarly, under normal gravity, the viability of 100 µg/mL TiO_2_-exposed cells was decreased to 78.64% as compared with that of TiO_2_-unexposed cells. 

In addition, the effects of TiO_2_ NPs exposure on apoptosis under normal gravity and simulated microgravity were explored. As shown in Figure 3A, under microgravity, TiO_2_ NPs exposure induced no obvious apoptosis at the low concentration of 25 μg/mL as compared with TiO_2_ NPs-unexposed cells. While the exposure concentration was increased to 100 μg/mL, the apoptosis rate was increased approximately by 10%, a bit lower than that of the counterparts under normal gravity. This result indicated that TiO_2_ NPs exposure induced a marginal cytotoxicity on A549 cells either under simulated microgravity or under normal gravity, with that of TiO_2_ NPs exposure less toxic under simulated microgravity relatively. The result was consistent with the result from CCK-8 assay (Figure 3B). The percentages of apoptotic cells including (Annexin V-FITC)+/PI+ and (Annexin V-FITC)+/PI− cells in each group were quantified in Figure 3C.

As demonstrated above, simulated microgravity has induced no significant cytotoxicity as compared with normal gravity. The exposure to a high concentration of TiO_2_ NPs (100 μg/mL) has elicited a slight decrease in cell viability accompanied with an increased apoptosis rate. Meanwhile, our results indicated that the condition of microgravity might help mitigate the cytotoxicity from TiO_2_ NPs exposure.

### 3.3. Effects of TiO_2_ NPs Exposure on Cell Migration and Cytoskeleton under Normal Gravity and Simulated Microgravity

The effects of TiO_2_ NPs exposure on A549 cell migration under the two conditions were evaluated by wound healing assay as described in Section 2. The result demonstrated that without TiO_2_ NPs exposure, the straight-line gap of cells under normal gravity was remarkably repopulated by A549 cells at 48 h since cell scratch was created, indicating a good motility of A549 cells under the normal culture conditions. Meanwhile, under simulated microgravity, a slightly lowered migration of A549 cells was exhibited by the smaller straight-line gap closure of cells. With 48 h of TiO_2_ NPs exposure under normal gravity, cell migration was decreased significantly in 25 µg/mL TiO_2_ NPs-exposed group. When the exposure concentration was increased to 100 µg/mL, a more obvious inhibition on cell migration was exhibited. Under simulated microgravity, the decrease in cell migration was relatively mild especially in 100 µg/mL TiO_2_ NPs-exposed group (Figure 4A). Therefore, our results demonstrated that simulated microgravity could decrease cell migration compared with normal gravity. TiO_2_ NPs exposure could inhibit cell migration both under the two different gravity conditions but to different extents. Quantitative analysis of relative cell wound closure rates was shown in Figure 4B.

The cytoskeleton is highly associated with cell migration. Recent studies also have found that the cytoskeleton may play an important role as the starting element in gravity sensing [64]. The cytoskeleton has been reported to be changed under microgravity or exposed to nanomaterials [38,65]. Herein, the effect of TiO_2_ NPs on cytoskeleton was explored under the two different gravity conditions. The observation towards rhodamine phalloidin-stained F-actin in the cultured cells under normal gravity distinctly showed the well-organized F-actin bundles that orderly radiated from perinuclear area to the cell periphery throughout cytoplasm. It seemed that the exposure to 25 µg/mL TiO_2_ NPs disturbed cytoskeleton and induced the condensation of cytoskeleton. When the exposure concentration of TiO_2_ NPs was increased to 100 µg/mL, actin filaments were disorganized, shortened and discontinuous (Figure 5). Meanwhile, the exposure to simulated microgravity tended to induce a weaken actin network as compared with that under normal gravity. When exposed to 25 µg/mL TiO_2_ NPs under simulated microgravity, the cytoskeleton changed more obviously with shrank and discontinuous F-actin. When the exposure concentration was increased to 100 µg/mL, F-actin density decreased and network structures were lost.

### 3.4. Effects of TiO_2_ NPs Exposure on Cell Adhesion-Associated Molecules under Normal Gravity and Simulated Microgravity 

Cell migration and cytoskeleton changes are considered to be tightly associated with cell adhesion molecules [63,66]. On the basis that TiO_2_ NPs exposure affected cell migration and cytoskeleton distribution under normal gravity and simulated microgravity, cell adhesion-associated molecules were explored. E-cadherin, as a cell adhesion molecule, is essential for maintaining the formation of adhesion links between adjacent homologous cells [67]. Integrins, on the other hand, transmit exogenous signals into the cell interior by mediating extracellular matrix (ECM) and focal adhesion (FA) [68]. The FA complexes include paxillin, vinculin, α-actinin, FAK, and other adhesion related molecules. Integrin β1, as a stress receptor in cellular response to external forces, is highly expressed in A549 cells, and it has been reported to be indispensable in mediating the cell adhesion to extracellular matrix [69,70].

In this study, to clarify the participation of adhesion-associated molecules, first, the spatial distribution and expression level of vinculin were explored after cells were exposed to different concentrations of TiO_2_ NPs under normal gravity or simulated microgravity for 48 h. The results from immunofluorescence analysis demonstrated that no obvious alteration in vinculin distribution was observed in TiO_2_ NPs-exposed cells no matter under normal gravity or simulated microgravity. In all the groups of cells, vinculin was distributed sparsely in the cytoplasm with scattered the formation of specific vinculin dots. The corresponding analysis of relative fluorescence intensity also exhibited no increase in the expression level of vinculin (Figure 6A,B). According to Western blot analysis, there was no significant difference in vinculin expression among all the cell groups of different treatments (Figure 6C).

Then, the expressions of E-cadherin and integrin β1 in each group were examined. The results from immunofluorescence staining of E-cadherin and integrin β1demonstrated that there was no significant difference in the expressions of E-cadherin (Figure 7A,B) and integrin β1 (Figure 8A,B) in the cells of different TiO_2_ NPs-exposed cells under simulated microgravity compared with normal gravity. These results also confirmed by the results of Western blot analysis (Figure 7C and Figure 8C).

Last, we studied the expression of FAK in TiO_2_ NPs-exposed cells under normal gravity and simulated microgravity. FAK, an important kinase linking to ECM and cytoskeleton, has been widely reported to involve in migration [71]. Our present study showed that the condition of simulated microgravity did not decrease the expression of FAK in TiO_2_ NPs-unexposed cells. When the exposure concentration of TiO_2_ NPs was 100 µg/mL, the expression of FAK decreased significantly both under normal gravity and simulated microgravity (Figure 9A,B).

As For FAK activation, the autophosphorylation of Tyr397 is critical for stimulating cell migration [72], and increased phosphorylation of Tyr397 leads to an increased cell migration. Herein, phospho-FAK (p-FAK) (Tyr397) was detected. Interestingly, the exposure to simulated microgravity induced a significant decrease in p-FAK (Tyr397) level as compared with normal gravity, which may account for the decreased cell migration as observed in the wound healing assay at least partially. When exposed to TiO_2_ NPs, p-FAK (Tyr397) levels were invariably decreased under both the gravity conditions, also correspondingly to the results of wound healing assay (Figure 9C,D). Our results indicated that the decreased level of p-FAK (Tyr397) corresponded to the decreased ability of cell migration. It infers that FAK greatly participated in cell migration when cells were exposed to simulated microgravity or TiO_2_ NPs.

Taken together, the disorganization of F-actin network and weakened FAK activation might be responsible for the impaired migrating capability of A549 cells when exposed to simulated microgravity or TiO_2_ NPs.

### 3.5. ICP-OES Determination of Cellular Uptake of TiO_2_ NPs 

Considering the possibility that the different cellular uptake of TiO_2_ NPs induced the different cell responses as demonstrated above, we determined the intracellular Ti concentrations in TiO_2_ NPs–exposed cells under normal gravity or simulated microgravity using ICP-OES.

As shown in Figure 10, no matter under normal gravity or simulated microgravity, the intracellular Ti concentration was invariably increased dependently on the exposure concentrations of TiO_2_ NPs. However, on the whole, the cellular uptake of TiO_2_ NPs in normal gravity-exposed cells was much more significant than that of simulated microgravity-exposed counterparts. This result indicated that A549 cells internalized more amount of TiO_2_ NPs under normal gravity than that under simulated microgravity, which may provide a possible explanation for the relatively lower cytotoxicity induced by the same concentration of TiO_2_ NPs exposure under simulated microgravity as compared with that under normal gravity.

## 4. Discussion

Since the great cost of astronaut flights in orbit doubtlessly imposes a limitation on the expanded scientific exploration especially towards life sciences, a range of microgravity-simulating devices have been developed for ground-based investigation, including the clinostat and the rotating wall vessel bioreactor [73]. Among them, the use of RPM in the laboratory setting is considered to be a useful tool for obtaining the simulated microgravity, advantageous for experimental research of cells in response to weightlessness [21,63,71]. Therefore, we used the RPM to create a simulated microgravity condition for our experiments.

In recent years, many researchers have studied the behaviors of cell proliferation, apoptosis and migration when cells are exposed to microgravity or TiO_2_ NPs. To date, there was no concerted conclusion on the effect of simulated microgravity on cell proliferation and apoptosis. The reports about promotion [25], inhibition [29] or no effect on cell proliferation under simulated microgravity all exist [74]. Our present study found that simulated microgravity had no obvious effect on A549 cell proliferation within 48 h. This result is similar to the previously reported studies that no difference in cell proliferation or apoptosis in microgravity-exposed and normal gravity-exposed human keratinocytes within 24 h [75,76]. As for TiO_2_ NPs exposure, no decrease in cell viability was detected within 24 h even when the exposure concentration was increased to 100 µg/mL. When the exposure time was extended to 48 h, a slight cytotoxicity was exhibited in 100 µg/mL TiO_2_ NPs-exposed cells. Aueviriyavit et al. [77] treated A549 cells with different concentrations (25-100 µg/mL) of TiO_2_ NPs (~25 nm) for 24 h, and no impact on cell viability was demonstrated. Mao et al. [78] incubated GC-2 and TM4 cells with TiO_2_ NPs (~21 nm, sphere, 100 µg/mL) for 24 h, only a slight cytotoxicity was induced. All of these reports indicated that the potential cytotoxicity of TiO_2_ NPs was marginal. Moreover, our present study suggested that the cytotoxicity of TiO_2_ NPs was still slight even under simulated microgravity. Similar to our results, Hekmat et al. [79] found that neither simulated microgravity (2D clinostat, 20 rpm) or the exposure of TiO_2_ NPs (<10 nm, 0–400 µM) did not affect the viability of MDA-MB-231 human breast cancer cells within 48 h, but the exposure of cells to TiO_2_ NPs under simulated microgravity led to a decreased cell viability.

In the present study, A549 cell migration was decreased when exposed to simulated microgravity as compared with the counterparts under normal gravity. Chang et al. [80] also reported that simulated microgravity reduced the metastasis potential of A549 cells by altering the expression of MKI67 and MMP2. Spaceflight studies have also found that microgravity slows the motility of the human monocyte J-111 [38]. In addition, under the normal gravity, the exposure of TiO_2_ NPs inhibited cell migration, especially at the high concentration of 100 µg/mL, suggesting the motility-decreasing effect of TiO_2_ NPs exposure itself. Mao et al. [78] also reported that the incubation of GC-2 cells with TiO_2_ NPs (100 µg/mL) significantly reduced the number of migrating cells, consistently with our results. However, under simulated microgravity, the migration of TiO_2_ NPs-exposed cells was inhibited as compared with that of TiO_2_ NPs-unexposed cells. When exposed to 100 µg/mL TiO_2_ NPs, a bigger inhibition of cell migration appeared under the normal gravity as compared with that under simulated microgravity. This phenomenon might be explained from the aspects of F-actin alteration as well as the cellular uptake of TiO_2_ NPs under normal gravity and simulated microgravity. 

As for F-actin, we found that F-actin of cells changed significantly under simulated microgravity with shortened microfilaments and decreased stress fibers while normal gravity-exposed cells exhibited a well-organized F-actin network, which was similar to the others’ results [38,71]. Meloni et al. [38] found the F-actin network in J-111 cells in-flight for 24 h rearranged, and the density of the filamentous biopolymer was significantly reduced. Similarly, Shi et al. [64] observed actin rearrangement in HUVECs cytoskeleton after 24 h simulated microgravity (2D-RWV, clinorotation at 30 rpm). Moreover, about 80% reduction in F-actin filaments were observed in the cells compared with normal gravity-treated cells. However, Hybel et al. [81] observed that an increased formation of lamellipodia and filopodia in prostate cancer cells line PC-3 under simulated microgravity (RPM, rotational acceleration was 20 °/s^2^) for 3 days and 5 days. Mann et al. [82] also found the same phenomenon that lamellipodia and filopodia were detected in RPM (60 °/s and 75 °/s) - exposed human foetal osteoblasts for 7 d and 14 d. The phenomenon of increased cellular stress fibers may be due to cell compensation for adhesion and changes in cell morphology to resist simulated microgravity conditions [81]. In brief, simulated microgravity-induced changes in F-actin depend on cell type or the length of exposure time. Upon TiO_2_ NPs incubation, the microfilaments of cells were disturbed obviously. The higher concentration of TiO_2_ NPs exposure, the more damage to F-actin network. The studies by others also reported the damage of TiO_2_ NPs to cytoskeleton [78]. Changes in F-actin may affect the formation of focal adhesion. To date, one article has mentioned the changes of focal adhesions during flight experiments [38]. In this study, 24 h after flight changed the localization of the vinculin significantly. The orientation of vinculin spots was no longer radial, but parallel to the cell membrane. Both real spatial and simulated microgravity experiments have shown the reduced J-111 cell migration during exposure to microgravity, which was considered to be associated with changes in cytoskeletal structure and focal adhesion plaques [38,65]. Although no alterations in the expression and distribution of vinculin were detectable in our results, cytoskeletal filaments were indeed disturbed with the deactivation of focal adhesion kinase (FAK).

As a signal protein of focal adhesion, FAK is a nonreceptor tyrosine kinase that plays an important role in cell migration. FAK has been shown to direct cellular responses by controlling lamellipodial protrusions and cell migration [83]. Studies have shown that FAK can promote breast cancer cell migration, and blocking or consuming FAK can impair cell migration ability [84,85]. Inhibition of FAK signaling in focal adhesions decreases cell motility [86,87]. Tan et al. [88] found that simulated microgravity using clinostat inhibited focal adhesions, leading to inhibition of FAK, RhoA and mTORC1 pathways. 

Phosphorylation of FAK at Tyr397 has been considered to be important for cell migration [72,85]. Nie et al. [89] found that inhibition of FAK phosphorylation by RGDS tetrapeptide significantly inhibited the adhesion, chemotaxis and invasion of A549 cells, indicating that FAK expression and activity regulate and change cell biological behaviors. Experiments have shown that the autophosphorylation of FAK at Tyr397 is critical to its ability to stimulate HUVEC cell migration [72], and increased phosphorylation of FAK at Tyr397 leads to an increased migration. Imaizumi et al. [90] believed that increased levels of protein tyrosine (include FAK), including phosphorylation, could inhibit cell adhesion and promote the migration, invasion and metastasis of lung cancer cells. In addition, Al-ghabkari et al. [91] inhibited the migration of human embryonal rhabdomyosarcoma cells by attenuating FAK phosphorylation at Tyr397 using a FAK inhibitor of PF-562271. All these data suggest the important role of FAK (Tyr397) in cell migration. Consistently with these reports, our results demonstrated that the level of p-FAK (Tyr397) was significantly reduced in simulated microgravity-exposed cells as compared with the normal gravity-exposed cells, which corresponds to the decreased cell migration under simulated microgravity. Upon TiO_2_ NPs incubation, cell migration was further reduced under both gravity conditions, while the level of p-FAK (Tyr397) was declined correspondingly. When the exposure concentration of TiO_2_ NPs was elevated to 100 µg/mL, even the total expression of FAK was obviously declined, suggesting the deactivation of FAK was involved.

When cells were exposed to TiO_2_ NPs, the different gravity conditions might impact the cellular uptake of TiO_2_ NPs, which finally affected cell behaviors. As discussed above, with the same exposure concentration of TiO_2_ NPs, the decreased cell viability, the increased apoptosis and the inhibited cell migration were a little milder under simulated microgravity than that under normal gravity. Meanwhile, the results from ICP-OES experiment demonstrated that the uptake of TiO_2_ NPs by A549 cells at the same concentration (100 µg/mL) under simulated microgravity was much lower than that under normal gravity. Considering the cytotoxicity of TiO_2_ NPs although it is not obvious, the lower cellular uptake under simulated microgravity might contribute to the observed lower toxicity relatively to that under normal gravity, at least partially. 

## 5. Conclusions 

In summary, our present results have demonstrated the exposure to simulated microgravity did not decrease cell viability but inhibited cell migration obviously with disturbed F-actin network. TiO_2_ NPs exposure induced a subtle cytotoxicity under both the two gravity conditions. However, with the same exposure concentration of TiO_2_ NPs, the decreased cell viability, the increased apoptosis and the inhibited cell migration were a little milder under simulated microgravity than that under normal gravity. This might be attributed to the lower cellular uptake of TiO_2_ NPs under simulated microgravity at least partially. Moreover, the decreased phosphorylation of FAK at Tyr397 could account for the lower cell migration from the aspect of a biological mechanism. Our study has provided some tentative information on the effects of TiO_2_ NPs exposure on cell behaviors under simulated microgravity.

## Figures and Tables

**Figure 1 nanomaterials-12-01879-f001:**
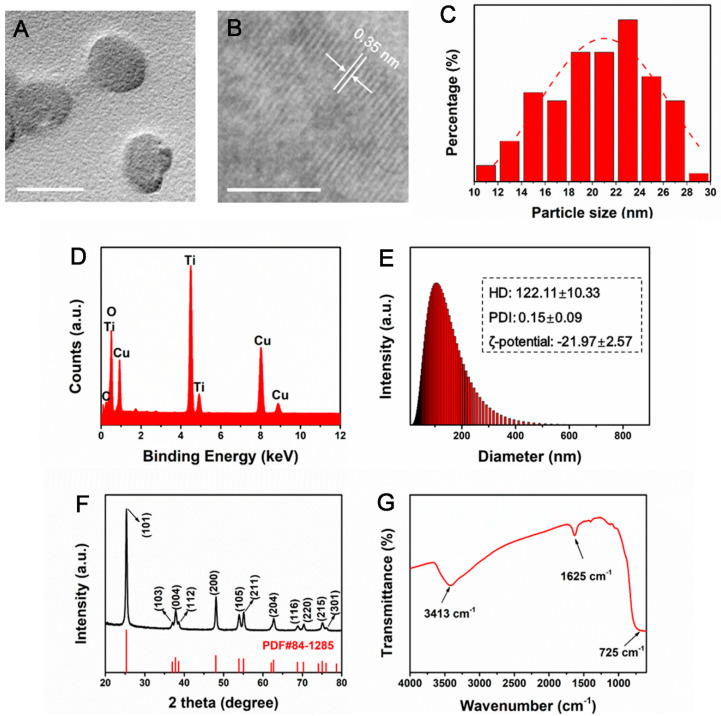
Physicochemical characterization of TiO_2_ NPs. (**A**) The representative TEM image (scale bar: 20 nm). (**B**) HRTEM image (scale bar: 5 nm). (**C**) Particle size distribution of TiO_2_ NPs. (**D**) Energy dispersive spectroscopy (EDS). (**E**) DLS measurements showing size distribution, HD, PDI and zeta potential. (**F**) Diffraction pattern showing crystalline structure. (**G**) FT-IR.

**Figure 2 nanomaterials-12-01879-f002:**
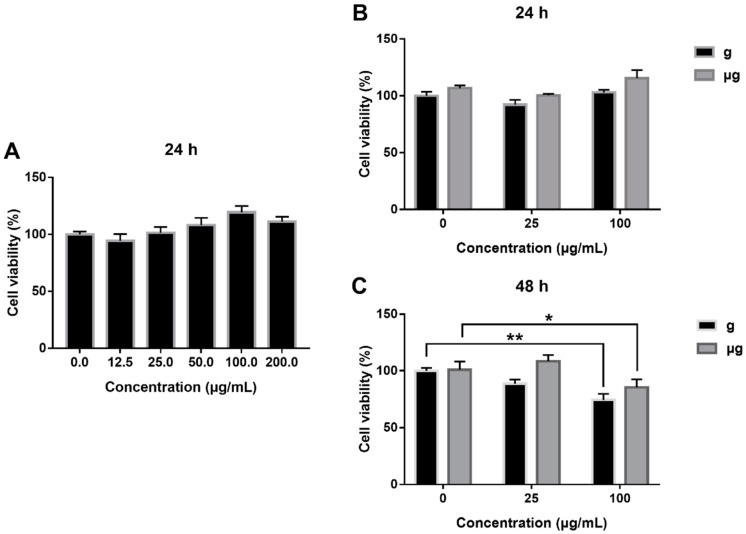
Effect of TiO_2_ NPs exposure on A549 cell viability. A549 cells were treated with TiO_2_ NPs (0, 12.5, 25, 50,100 and 200 µg/mL) for 24 h under normal gravity (**A**). And cells were treated with TiO_2_ NPs (0, 25 and 100 µg/mL) for 24 h (**B**) or 48 h (**C**) under normal gravity (g group) or simulated microgravity (µg group). Cell viability was measured using Cell Counting Kit-8 assay. * *p* < 0.05, ** *p* < 0.01.

**Figure 3 nanomaterials-12-01879-f003:**
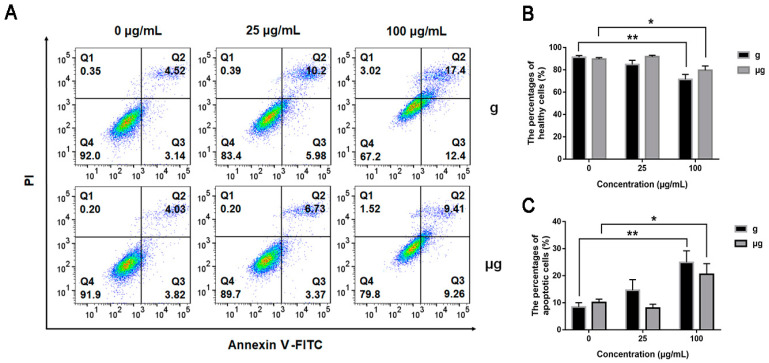
Cytotoxic effects of TiO_2_ NPs exposure under normal gravity and simulated microgravity. A549 cells were treated with TiO_2_ NPs (0, 25 and 100 µg/mL) for 48 h under normal gravity (g group) or simulated microgravity (µg group). (**A**) Apoptosis analysis by flow cytometry using Annexin V-FITC/PI staining kit after 48 h of TiO_2_ NPs exposure. Representative results of flow cytometry analysis were shown. The percentages of healthy cells ((Annexin V-FITC)−/PI−) (**B**) and apoptotic cells ((Annexin V-FITC) +/PI + plus (Annexin V-FITC)+/PI−) (**C**) were presented. (Annexin V-FITC)+/PI+: late apoptotic/necrotic cells, (Annexin V-FITC) -/PI +: necrotic cells, (Annexin V-FITC)+/PI−: early apoptotic cells, (Annexin V-FITC)−/PI−: viable cells. The histograms represent the mean ± SD of at least three independent experiments. * *p* < 0.05, ** *p* < 0.01.

**Figure 4 nanomaterials-12-01879-f004:**
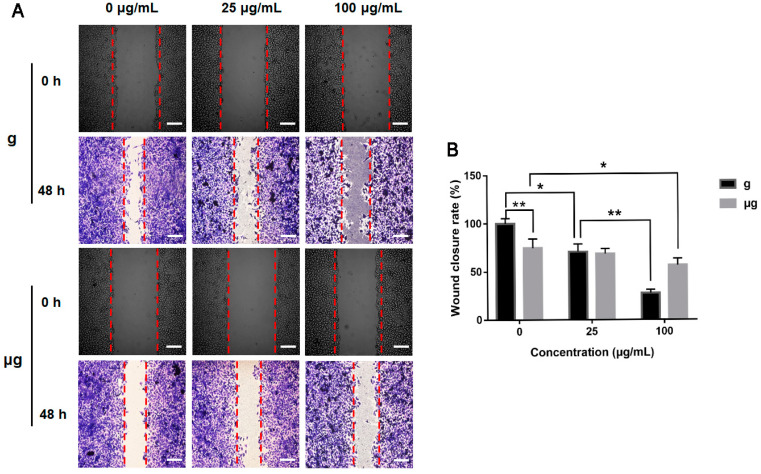
Effect of TiO_2_ NPs exposure on A549 cell migration under normal gravity and simulated microgravity. (**A**) A549 cells were treated with TiO_2_ NPs (0, 25 and 100 µg/mL) for 48 h under normal gravity (g group) and simulated microgravity (µg group), respectively. Images were taken at 0 and 48 h after scratching. The red dotted lines show the boundaries of the migratory A549 cells (scale bar: 100 µm). (**B**) Quantitative analysis of relative cell wound closure rates. The histograms represent the mean ± SD of at least three independent experiments. * *p* < 0.05, ** *p* < 0.01.

**Figure 5 nanomaterials-12-01879-f005:**
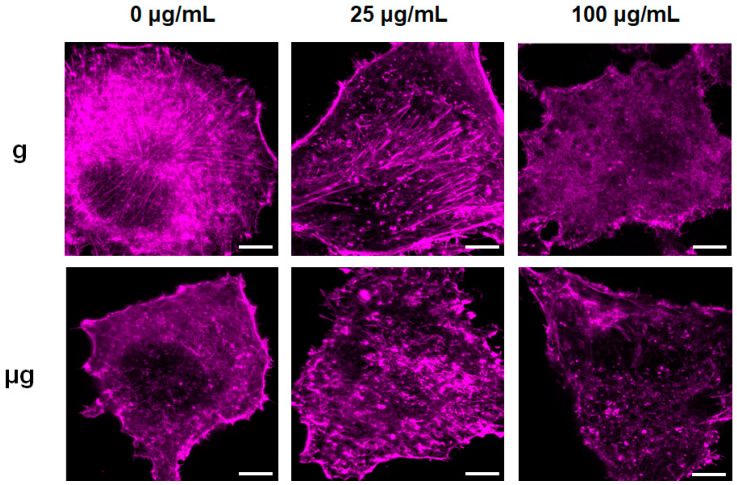
Observation of F-actin distribution by rhodamine phalloidin staining. A549 cells were treated with TiO_2_ NPs (0, 25 and 100 µg/mL) for 48 h under normal gravity (g group) and simulated microgravity (µg group), then cells were fixed and stained with Rhodamine Phalloidin, and representative fluorescent images were demonstrated here (pseudo color: purple). Scale bar: 5 µm.

**Figure 6 nanomaterials-12-01879-f006:**
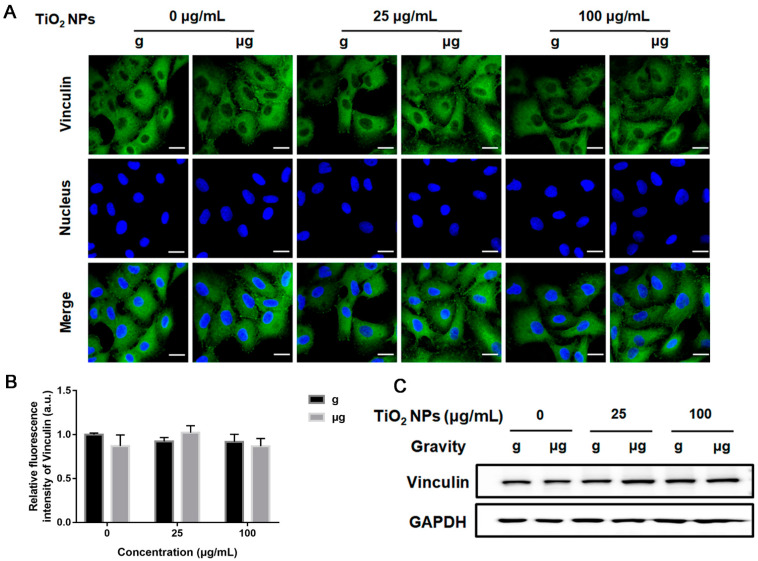
Effects of TiO_2_ NPs exposure on the spatial distribution and expression level of vinculin. A549 cells were treated with TiO_2_ NPs (0, 25 and 100 µg/mL) for 48 h under normal gravity (g group) and simulated microgravity (µg group). (**A**) The harvested cells were subjected to immunofluorescence staining for vinculin observation. DAPI was used to stain the nuclei (blue). The representative immunofluorescence images of cells with different treatments were demonstrated here (scale bar; 20 µm). (**B**) The quantitative analysis of fluorescence intensity. (**C**) The representative results of Western blot analysis showing the expression level of vinculin in A549 cells of different groups.

**Figure 7 nanomaterials-12-01879-f007:**
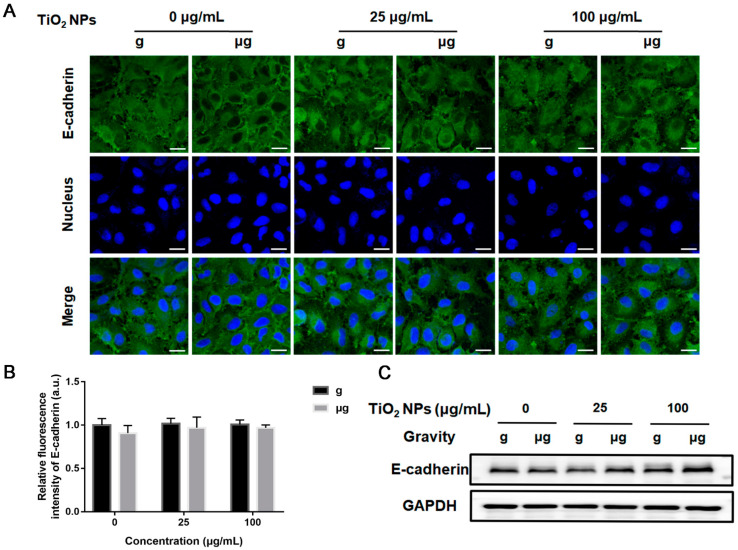
Effect of TiO_2_ NPs exposure on the expression level of E-cadherin. A549 cells were treated with TiO_2_ NPs (0, 25 and 100 µg/mL) for 48 h under normal gravity (g group) and simulated microgravity (µg group). (**A**) The harvested cells were subjected to immunofluorescence staining for E-cadherin observation using a confocal laser scanning microscope. DAPI was used to stain the nuclei. Representative immunofluorescence images were demonstrated here (scale bar; 20 µm). (**B**) The quantitative analysis of the fluorescence intensity of each group. (**C**) The representative Western blot results demonstrating the expression levels of E-cadherin in each group of A549 cells.

**Figure 8 nanomaterials-12-01879-f008:**
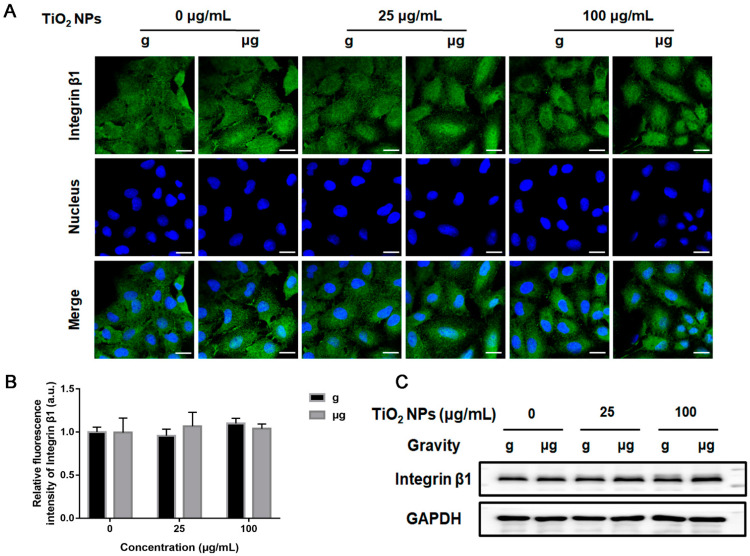
Effect of TiO_2_ NPs exposure on the expression level of integrin β1 upon different treatments. A549 cells were treated with TiO_2_ NPs (0, 25 and 100 µg/mL) for 48 h under normal gravity (g group) or simulated microgravity (µg group). (**A**) The harvested cells were immune-stained for integrin β1 observation. DAPI was used to stain the nuclei. Representative immunofluorescence images were demonstrated here (scale bar; 20 µm). (**B**) The quantitative analysis of fluorescence intensity of each group. (**C**) The representative results of Western blot analysis.

**Figure 9 nanomaterials-12-01879-f009:**
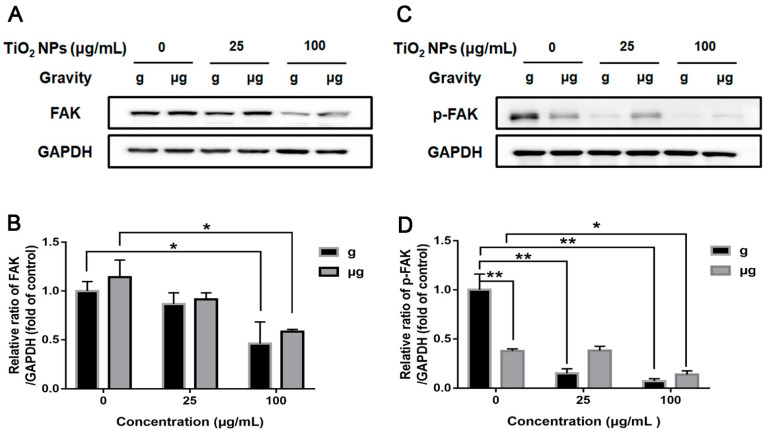
Effect of TiO_2_ NPs on the expression levels of FAK and p-FAK (Tyr397). A549 cells were treated with TiO_2_ NPs (0, 25 and 100 µg/mL) for 48 h under normal gravity (g group) or simulated microgravity (µg group). Representative results of Western blot analysis showed the protein expressions of FAK (**A**) and p-FAK (**B**) in A549 cells. Quantitative analysis of FAK (**C**) and p-FAK (**D**) were shown. * *p* < 0.05, ** *p* < 0.01.

**Figure 10 nanomaterials-12-01879-f010:**
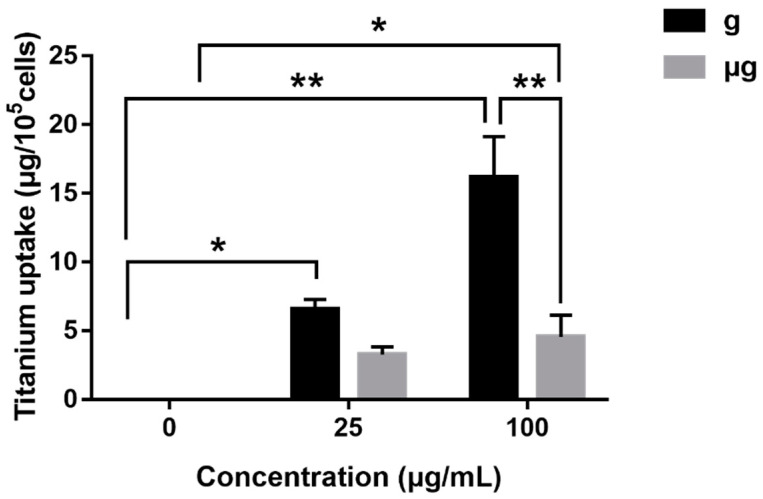
Cellular uptake of TiO_2_ NPs by A549 cells. A549 cells were treated with TiO_2_ NPs (0, 25 and 100 µg/mL) for 48 h under normal gravity (g group) and simulated microgravity (µg group). Ti content was determined by ICP-OES. A calibration curve was constructed using standard titanium dilutions of a reference solution and the correlation coefficient (R^2^) was 0.99996. All values were presented as mean ± SD (*n*= 3). * *p* < 0.05, ** *p* < 0.01.

## Data Availability

Not applicable.

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
