# Peer review of "Effects of Titanium Dioxide Nanoparticles on Cell Growth and Migration of A549 Cells under Simulated Microgravity"

_nanomaterials, 2022, doi:10.3390/nano12111879_

Round 1
Author Response
Thank you for your comments.
Reviewer 2 Report
In the manuscript I am reviewing autors investigated the behaviors of cells under simulated microgravity and also evaluated the possible toxicity of TiO2 NPs. The manuscript is extremely scientifically interesting and well written. The methods used are described in detail and the results are clearly presented to the reader.
I have two suggestions for changes:
- replace figure 1A with a better quality
- correct a typo - line 191: in a 37 ◦C incubator
Reviewer 3 Report
The current manuscript describes the impact of the combination of microgravity and TiO2-NP on A549 cells. This topic is relevant since astronauts can be exposed to this nanomaterial. Interestingly, both microgravity or TiO2-NP alone have an impact on cell cytoskeleton but the combination does not induce a synergistic impact. On the contrary, there is even no additional impact for some of the experiments presented. Overall, the conclusions are not highly novel but the manuscript is anyway of interest in the field and could be published after performing the following corrections and additions.
Major corrections:
Figure 6, 7 and 8 : Immunofluorescence images are not convincing for vinculin, E-cadherin and integrin-beta1. Higher magnifications are required to be able to confirm the conclusion drawn on Figure 6A. For integrin-beta1, it is even worst with a very low intensity of fluorescence signal that needs to be improved before publication.
Considering Ti uptake measurement, it is known that TiO2-NP are rock solid. So how can you be sure that your procedure is sufficient to completely mineralize these NPs into atoms? A proper control experiment on pure TiO2-NP is required to confirm this point.
Minor correction : Panel B and C are inverted in Figure 9.
Round 2
Reviewer 3 Report
After revision the manuscript is now suitable for publication.